# Choosing the Right Path for the Successful Storage of Seeds

**DOI:** 10.3390/plants12010072

**Published:** 2022-12-23

**Authors:** Magdalena Trusiak, Beata Patrycja Plitta-Michalak, Marcin Michalak

**Affiliations:** 1Department of Plant Physiology, Genetics and Biotechnology, University of Warmia and Mazury in Olsztyn, M. Oczapowskiego 1A, 10-721 Olsztyn, Poland; 2Department of Chemistry, University of Warmia and Mazury in Olsztyn, Plac Łódzki 4, 10-719 Olsztyn, Poland

**Keywords:** seed storage, desiccation, orthodox, recalcitrant, intermediate, exceptional species, cryopreservation

## Abstract

Seeds are the most commonly used source of storage material to preserve the genetic diversity of plants. However, prior to the deposition of seeds in gene banks, several questions need to be addressed. Here, we illustrate the scheme that can be used to ensure that the most optimal conditions are identified to enable the long-term storage of seeds. The main questions that need to be answered pertain to the production of viable seeds by plants, the availability of proper protocols for dormancy alleviation and germination, seed tolerance to desiccation and cold storage at −20 °C. Finally, it is very important to fully understand the capability or lack thereof for seeds or their explants to tolerate cryogenic conditions. The proper storage regimes for orthodox, intermediate and recalcitrant seeds are discussed.

## 1. Introduction

The global biodiversity crisis has become a problem over recent years, mainly due to human activity, and it is expected to continue. The causes can be direct including land use, pollution or climate change, and indirect, as demographic factors, as well as economic and government issues [1,2]. Currently, the increase in average year temperature and related changes in climate are undebatable. Indeed, we have been experiencing global climate warming since 1981—the rate of temperature increase has been 0.18 °C per decade, and the 9 years from 2013 through 2021 rank among the 10 warmest years on record (www.climate.gov/news-features/understanding-climate/climate-change-global-temperature, accessed on 28 July 2022). It has become clear that climate change entails the loss of biodiversity and impacts its organization at all levels, affecting genes, populations, species, ecosystems, and thus human livelihood [3,4,5,6,7]. Two general approaches exist for dealing with the consequences of climate change, which are an adaptation to adverse conditions by adjustment to high temperatures or mitigation of the consequences of a warming climate. In both approaches, however, conservation of the world’s biological resources and their diversity is crucial to enable future generations to thrive and adjust to altered climate conditions as well as counteract further degradation of ecosystems. Only by preserving diversified varieties of plants will we be able to secure all multiple and vastly diverse products and services provided by plants including food and medicine, timber and animal forage, as well as a regulation of water supply and carbon sequestration [4,8,9,10]. Although there are still few (0.2%) well-documented plant extinction examples, it has been estimated that 30–44% of land plants are threatened [9]. The understanding of the importance of biodiversity protection for future human prosperity has increased the number of efforts aiming at in situ actions such as expanding the protected areas and restoring degraded ecosystems [9]. Over the course of the last several decades, conservation activities away from the natural location of the plants (ex situ) have resulted in an increased number of gene banks worldwide and a rapid rise in the number of accessions that are preserved [11,12,13,14,15,16,17,18]. Gene banks provide a broad range of diversity for usage in breeding programs and research efforts, and they play a key role in disseminating seeds to farmers [19]. Lastly, they serve as an invaluable backup source of seeds in the event of a disaster [20]. Currently, there are approximately 1750 gene banks in existence which maintain more than 7.4 million accessions of plant genetic resources. Among these gene banks, we can distinguish approximately 130 large facilities that hold more than 10,000 accessions at each location. There are also substantial ex situ collections in botanical gardens, containing approximately 4 million accessions from over 80,000 species [20,21,22]. Importantly, not only are the crop varieties and their wild relatives currently preserved, but also wild species as well. Among the main largest-scale activities planned to enlarge the deposition of seeds was the operation of the Millennium Seed Bank: “Collect and conserve 10% of the world’s wild seed-bearing flora, principally from the drylands, by the year 2010”; currently, the aim has been expanded to 20% [23]. Another important initiative is the Consultative Group for International Agricultural Research (CGIAR) that operates gene banks preserving the collection of >700,000 crop seed accessions [11,19].

The hope and expectation that seed storage and gene banks can effectively contribute to the conservation of plant biodiversity in a time of complex global challenges relate to studies showing that seeds can remain viable for hundreds and even thousands of years [20,24,25]. *Silene stenophylla* Ledeb. buried in permanently frozen loess-ice deposits on the right bank of the lower Kolyma River for 32,000 years are the oldest known viable seeds that have been found [26]. However, these data are often burdened with error as the age of seeds was assessed by radiocarbon dating [27]. Nevertheless, the longest ongoing experiment and, in general, among the oldest trials in the world regarding seed longevity, was initiated in 1879 at Michigan State University by Dr. William James Beal. In this study, 20 bottles were buried underground on the MSU campus, with each containing 50 seeds from 21 different plant species. The intention was that a bottle would be dug up every 5 years but intervals were subsequently extended. The last published data of a 120-year storage period showed that 23 seeds of *Verbascum blattaria* L. and 2 seeds of *Verbascum* sp. germinated, which corresponded to 50% of total germination [28]. The last sample was taken in 2021 and the international community of seed researchers is still looking forward to a publication which will reveal seed germinability, as this experiment is a unique mix of seed storage in semi-natural conditions with carefully recorded data. Consequently, based on the natural features of seeds, the long-term security of seeds in cold storage facilities was established and projected.

The aim of this review is to provide a comprehensive view on current possibilities for the long-term preservation of seeds. In previous publications, information pertaining to how seeds should be stored based on their post-harvest physiology [29,30,31] or how to improve conditions for more successful cryostorage of extracted plant tissues has been described [21,32]. However, in preparation of a decision-making path for the successful storage of plant germplasm, we focused on seeds, from their collection until conventional and cryogenic storage (Figure 1). Importantly, our decision-making path is enriched with information on how to establish and choose the optimal conditions of storage when protocols of dormancy alleviation and seed germination are not available. Unfortunately, this is a frequent and very problematic situation, especially in the case of endangered and rare species originated from biodiversity hot spots. This important information and guidance is regularly omitted in other reviews, as the most commonly presented decision-making process is typically based primarily on the desiccation tolerance of seeds [29,30,33]. Since several recent studies showed that there is high probability to predict seed desiccation tolerance [34,35], it is plausible that efforts to preserve valuable seeds could be initiated and proceeded until the dormancy breaking and germination protocols are provided. Therefore, there is a need for an indication regarding how to proceed with such seeds. We have described the respective stages of a decision-making pathway in more detail, and also provide the most important information regarding the influence of condition of storage on seeds’ viability. Finally, we provide some information on future possibilities of improving both conventional and cryogenic storage of seeds. Collectively, all of this information provides indications for how to search for the solution if problems in seed storage are encountered.

## 2. Important Questions That Need to Be Addressed Prior to Seed Storage

The first question that needs to be answered is whether the species of interest produces viable seeds? This is an important question because even though there is the possibility to store other tissues, i.e., shoot tips, dormant buds, pollen, or vegetative clonal tissues, seeds are the first choice to protect biodiversity as each seed is an individual specimen containing a diverse genetic material that enables the protection of a wide gene pool [21,36,37,38,39,40]. Moreover, under appropriate conditions, seeds produce seedlings without the requirement of additional, costly and labor-intensive procedures, such as in vitro culture. Therefore, the higher plant accessions are usually stored in gene banks as seeds [40,41]. If the species of choice does not produce viable seeds that could be stored, other approaches to preserve the plant in gene banks are chosen, including storage of other organs and tissues mentioned above. The present article is focused on the preservation of seeds. We propose the decision-making path for the successful storage of plant germplasm, with a focus on seeds (Figure 1).

Numerous factors can influence the ability to successfully procure and use seeds to support plant reintroductions, including challenges with seed collection, storage, and germination [31,42]. Therefore, when seeds are available, and confirmed to be viable by, e.g., biochemical 2,3,5-tryphenyl tetrazolium chloride (TTC) test [43] and proved to be at good quality by, e.g., X-ray imaging [44,45], the next valid issue that requires an answer is the accessibility of the germination and dormancy breaking protocols if seeds become dormant upon maturation. Dormancy regulates germination through various physical and/or physiological means imposed by the seed coat, or within the embryo. Freshly collected seeds are considered to be dormant when they do not germinate within 4–6 weeks under conditions of sufficient moisture and suitable temperature [46,47]. Unfortunately, for many wild species, there are no such protocols. However, 50% to 90% of wild plant species produce dormant seeds, with specific dormancy traits driven by the geography of species’ occurrence, a form of growth, and genetic factors [47]. For example, more than 70% of the arctic-alpine plants produce dormant seeds, exhibiting mainly physiological dormancy, which evolved to prevent seed germination before or during cold autumn/winter seasons [46]. Unfortunately, the protocols for dormancy breaking require laborious, long-term procedures as seeds may require several months or years to lose their dormancy [47]. Therefore, specialized knowledge regarding the ecology and physiology of the species is a prerequisite. As a consequence, the accessibility to proper dormancy-breaking and germination protocols can be the bottleneck limiting conservation efforts, because this challenging question needs to be faced: How to proceed with the storage of seeds without proper protocols essential for testing seed viability before and after storage? Consequently, understanding seed dormancy and germination traits is critical to effective seed management and achieving planned restoration success [47]. In general, it is assumed that closely related species growing in similar conditions and climatic zones produce seeds of similar characteristics [48]. However, it should be remembered that this is not always a true assumption. For instance, the dormancy phenotype of *Avena fatua* L. seeds is determined 50% genetically, by 50% in the maternal environment, and therefore results in variation [49]. Additionally, the response of mature seeds of *Amaranthus retroflexus* L. to temperature and the light regime was affected by harvesting time as they varied in dormancy levels and requirement for after-ripening [50]. There was a difference in optimal germination temperature for seeds produced by *Vicia sativa* L. cultivars collected at a different elevations above sea level [51]. Species-specific dormancy alleviation protocols were reported for *Violaceae*, as some species required cold stratification while others responded to other procedures such as pretreatment with gibberellic acid and priming [42]. Thus, it is well known that dormancy alleviation and germination stimulation may vary between individual plants from one population, from year to year, and between sites of production which is attributed to differences in environmental factors before and during seed maturation [47,49,50,51]. Lack of knowledge about seed dormancy and germination requirements leads to failure in the regeneration of plantlets and seed wastage [47]. Therefore, if there is no species-specific information in the literature and on Royal Botanic Garden Kew’s Seed Information Database, the “move-along” experiment testing dormancy breakage and germination on a small batch of freshly collected seeds in conditions simulating a series of temperature regimes mirroring natural conditions in any bioregion is proposed [47,52,53]. In addition, the exact seed bank procedure for testing dormancy breakage and germination conditions is also established [29].

## 3. How to Successfully Store Seeds for a Long Time? The Role of Seed Moisture Content (MC) and Low Temperature in an Extension of Seed Longevity

Another important issue that must be tackled prior to the deposition of seeds in storage is the determination of their desiccation tolerance. Since the 1970s, the most common and popular classification of seeds based on their post-harvest physiology has been used [54]. Seeds were divided into two categories according to their resistance to desiccation which is defined as a reduction in their MC that is not detrimental to viability. Orthodox seeds are capable of tolerating desiccation below 5% of their MC and resume metabolism when rehydrated. This unusual feature of life objects is limited to microbes, invertebrates, resurrection plants, fungi, and seeds [55,56,57,58]. Mature orthodox seeds are developmentally arrested, metabolically quiescent, and dry. Thus, seeds shed or harvested at a MC of 20% or below are very likely to be orthodox [30,58,59]. In contrast, recalcitrant seeds are frequently shed from the mother plant at a MC of above 60% that may trigger germination [60]. However, these seeds are sensitive to desiccation as they do not tolerate drying below 20–40% of MC [30,54]. The distinction has been used as a management criterion for storage in gene banks [61,62]. Further exploration of seed behavior raised the awareness that this orthodox-recalcitrant paradigm is too restrictive, and in consequence, this binary division was extended by adding an intermediate category [63]. As the name suggests, this category of seeds exhibits to various extents some features of both orthodox and recalcitrant seeds. As a result, this seed category is less uniform in relative comparison to other categories. Indeed, these seeds (1) tolerate desiccation to a MC of 10%, although it is not so low as in the case of orthodox seeds. They are (2) sensitive to storage at sub-zero temperatures and (3) are characterized by a short lifespan regardless of the effort aiming to improve its extension by lowering the MC and temperature [62]. Among the explanations of short-lived behavior is that seeds are dispersed without a complete metabolic shut-down, as in the case of desiccation tolerant *Poincianella pluviosa* (DC.) L. P. Queiroz [59]. Another reason may be related to a lack of endosperm and an energy reservoir, as in the case of *Populus* sp. seeds [64]. Nevertheless, this categorization is useful for the decision-making process in seed management, however, it should be kept in mind that seeds exhibit a wide spectrum of responses to desiccation rather than the rigid ones defined by the categories [57,62]. Therefore, because the thresholds among orthodox, intermediate, and recalcitrant seed behavior are not always well defined, efforts to allocate seeds in a particular category may lead to varied conclusions. For example, in the case of *Corylus avellana* L., its seeds were already classified as recalcitrant, intermediate and orthodox [65,66,67].

Once the tolerance to desiccation is tested, the next step in the process is the determination of the proper temperature of storage. For decades, it has been assumed that lowering both the seed MC and temperature of storage would increase their lifespan. The first rules aiding in the establishment of the seed storage regime were “the rules of thumb” [68,69]. The following rules allow for an assessment of the effect of MC and temperature on seed storage, stating that (1) for every 1% decrease in seed MC, the lifespan of seeds doubles. This rule is applicable when seed MC ranges between 5% and 14%. (2) For every 5 °C (10 °F) decrease in the storage temperature, the life of the seed doubles when the temperature ranges from 0 to 50 °C. (3) Optimum seed storage is achieved when the sum of percent relative humidity (RH) in the storage environment and the storage temperature in F sum up to a hundred, but the contribution from temperature should not exceed 50 °F. In the 1980s and 1990s, the scientific debate regarding the most proper MC for the long-term storage of seeds continued. According to R.H. Ellis and collaborators, seed longevity increases as seeds are progressively dried and temperature and MC have an independent impact [70,71,72,73]. Therefore, seeds were suggested to be stored at an extremely low level of MC close to 10–12% of RH achieved during drying at 20 °C, regardless of the storage temperature. Conversely, Vertucci and collaborators [74,75,76] claimed that over-drying of seeds could be deleterious to their viability. The optimal seed MC for storage depends on the temperature of storage, and it decreases as temperature increases. Therefore, the optimal RH for seed storage at ambient temperature should be approximately 20–25% RH [76]. On the other hand, for cryogenically stored seeds (temperature approximately −150 °C), the RH should be higher (i.e., for pea seed, this is approximately 0.17 g H_2_O g^−1^ dry mass, which is equivalent to 70% RH) [75]. Nevertheless, desiccation tolerance and optimal temperature of storage, if not available in databases (i.e., Seed Information Database (SID; http://data.kew.org/sid/, accessed on 4 October 2022), need to be established experimentally. When there is a limited number of available seeds, desiccation tolerance can be assessed by a so called “100-seeds test”. This test was initially developed for palm [77], and was further successfully adapted to 91 native Caribbean woody species [78]. Additionally, if there is no reference to information pertaining to the storage of seeds for a particular species, the desiccation tolerance and category of seeds could be predicted by models that are based on seeds’ trait such as seed coat ratio and seed mass [35,79]. Additional research on 17,539 species showed that desiccation tolerance can be anticipated based on taxonomy, species traits, location and climate data. Importantly, the most important predictor variables were the response of relatives to desiccation, seed mass and annual precipitation. Three models with varying degrees of success rates for identifying the desiccation-sensitive species were constructed based upon these variables. Specifically, there was an 89% success rate for the genus-level model, 79% for the family-level model and 60% for the order-level model [48].

If information exists on how to store seeds of particular or closely related species, such seeds should be collected when ripe, preferably shed, and only slight subsequent surface desiccation is recommended prior to storage at low temperature. The temperature level is dependent on the climate conditions where maternal plants grow and the chosen storage temperature should be close to the average temperature for the winter season. However, such conditions of storage should be applied only for short-term storage until a proven method is developed. For example, a method can be determined based upon the step-by-step approach [29,30], in which the desiccation tolerance to 10–12% MC is first evaluated and then further to 5%, which subsequently enables the possibility to store seeds for 3 months at −20 °C. If seeds survive desiccation to 5% of MC and storage at −20 °C, it is very likely that they belong to the orthodox category. For other seeds, experimental data need to be collected as they require non-conventional storage methods [80]. The experimental path to determine intermediate or recalcitrant seeds has been proposed [30,81]. It includes storage at +5 and −20 °C and regular viability testing for a 5-year period. By using this approach, it can be determined whether seeds belong to the intermediate short-lived or intermediate freeze-sensitive categories. Examples of orthodox, intermediate, and recalcitrant seeds storage conditions, and related issues are presented in the sections below.

## 4. Conventional Storage of Seeds

The term “conventional storage” means that seeds are treated in a defined way. Specifically, they are desiccated to 10–25% of RH in a recommended drying environment of 5–20 °C, which corresponds to approximately 5% of MC [61]. According to the established protocol [30], for the short-term storage of dry seeds (<18 months), a temperature between 0 and 5 °C is sufficient to maintain their viability. For longer periods of storage, seeds should be stored at −18 to −20 °C. However, in Mediterranean gene banks, seeds of native species are stored at −25 °C [82,83].

### 4.1. Conventional Storage of Orthodox Seeds

Multiple reports have described the successful conventional storage of orthodox seeds [80,84,85,86,87,88,89,90,91]. In particular, results obtained in long-term (years and decades) studies are of high importance as they are proof of concept and provide actual information on the longevity of orthodox seeds. For instance, 16 Nordic agricultural and horticultural crops, each represented by two or three cultivars, were stored under ultra-dried conditions at a MC of 3–5% a temperature of −3.5 ± 0.2 °C for a 30-year period. The aforementioned seeds remained highly viable for 20–25 years. However, after 30 years of storage, the median value of germinability for all tested seed lots decreased to 80%, with the highest germinability decline (up to 49%) observed for *Secale cereale* L. found in seed lots with the highest initial MC [88]. Another study showed that the longevity of desiccation-tolerant seeds of 28 plants from 7 families conventionally stored at −18 °C and MC of 5 ± 2% varied from 20.41 years (*Arachis hypogea* L.) to 500 years (for *Avena sativa* L. and *Triticum aestivum* L.) [92]. During one of the most complex studies, 42,000 seed accessions representing 276 species were quantified after storage at 5 °C, followed by the storage of a select 178 species at −18 °C [93]. The MC of the investigated seeds ranged between 4% and 8%. This experiment showed that the time to reduce germination to 50% (P_50_) ranged among species from <13 years to an extrapolated estimate of >450 years, and the median P_50_ was 54 years. Some plant families possessed characteristically short-lived (i.e., Apiaceae and Brassicaceae) or long-lived (e.g., Malvaceae and Chenopodiaceae) seeds. Although data from that study support the statement that some species tend to survive longer than others in the seed bank, the information on the attributes of seeds affecting their storage performance is still incomplete. Importantly, a meta-analysis of germination data indicated that predicted germinability was often higher than actual values that were experimentally obtained [87]. Therefore, further research which focuses on obtaining more accurate predictions of orthodox seed longevity based upon the testing of new markers of seed viability and quality in real-time appear warranted and necessary [91,94,95,96,97,98,99].

### 4.2. Conventional Storage of Intermediate Seeds

The first definition of the intermediate category states that among the main features of these seeds is that they can be injured by low temperatures in a dry state [63]. Indeed, the viability of coffee seeds (*Coffea arabica* L.) stored at cool and sub-zero temperatures and at a low MC was shown to be lower in comparison to seeds stored at a higher MC and temperatures. In accordance with these observations, seeds of *Citrus* species behaved similarly. Moreover, in the case of *Fagus sylvatica* L. and *F. crenata* L., seeds of both species do not withstand desiccation below 7.6%, thus they are not orthodox, but they can be stored in and optimal temperature range of +10 to −20 °C [100]. These aforementioned seeds were classified as intermediate, as well as other oil-rich seeds such as *Coffea arabica*, *Citrus* sp., and *Corylus* sp. [101]. However, there are also short-lived seeds in the intermediate category, i.e., *Salix* sp. [102] or *Populus* sp. [103,104], that do not follow typical orthodox or recalcitrant seed storage behavior. Lastly, there are also seeds which do not fit the classification as recalcitrant or orthodox, that have been placed in the intermediate category. As a consequence, this group is highly heterogenous [62]. Nevertheless, the longevity of intermediate seeds increases with drying and cooling (as with orthodox seeds), but seeds still age rapidly during conventional storage and will die within approximately 5 years ([62], saveplants.org/best-practices/difference-between-orthodox-intermediate-and-recalcitrant-seed/ accessed on 28 July 2022). Relevant information regarding the storage behavior of intermediate seeds for their conservation appears to have gained importance since recent research showed that more seeds, particularly from the world’s biodiversity hotspots, should be classified as intermediate [80,81,105]. It has also been claimed that desiccation and temperature behavior during storage lasting longer than 2 years should be studied comparatively to properly determine the intermediate category [81].

### 4.3. Conventional Storage of Recalcitrant Species

Recalcitrant seeds do not withstand conditions of conventional storage due to mechanical damage, metabolism-induced damage and macromolecular denaturation. Importantly, recalcitrant storage behaviors are more prevalent in woody plant species [31]. Therefore, the most practical way to extend the storage life of intact recalcitrant seeds is by storing them at the lowest possible temperature that does not cause damages as mentioned above [106]. However, the lowest applicable temperature must be experimentally tested. For instance, recalcitrant seeds from the temperate zone can be stored for a short time at the temperature range between 0 and −10 °C, i.e., *Quercus robur* L. [107,108,109] and *Acer pseudoplatanus* L. [110], while short-term storage of tropical recalcitrant seeds, i.e., *Hopea hainanensis* Merr. at a high MC of ~33% and a much higher temperature of 15 and 20 °C was reported [111].

### 4.4. Conventional Storage of Woody Plant Species in Gene Banks

In recent decades, the importance of preservation of tree species in gene banks has become indisputable as approximately 30% of tree species are threatened with extinction and at least 142 tree species are recorded as extinct in the wild (https://www.bgci.org/resources/bgci-tools-and-resources/state-of-the-worlds-trees/, accessed on 20 October 2022). As a result, new gene banks that are focused mostly on preservation of forest tree species were established, e.g., Kostrzyca Forest Gene Bank in Poland [17,112]. Another example of conservation efforts was the UK National Tree Seed Project (UKNTSP), that was initiated in 2013 by Royal Botanic Garden Kew and completed in April 2018. This project included a collection and conventional storage of 10 million seeds from 60 woody species native for the UK (https://www.kew.org/read-and-watch/10-million-seeds-national-seed-project, accessed on 20 October 2022). Additionally, seeds of 131 threatened woody plant species collected worldwide are preserved at −20 °C in Millennium Seed Bank (Appendix A based on [90]. Moreover, in The World Agroforestry Center (ICRAF) in Nairobi, 5800 seed accessions, or unique seed samples, representing 189 tree species, are conserved (https://forestsnews.cifor.org/65471/world-agroforestry-gene-bank-germinates-a-future-for-healthy-ecosystems?fnl=en, accessed on 20 October 2022).

## 5. Cryogenic Storage of Seeds

Cryogenic storage offers a long-term method of plant germplasm preservation that, in principle, overcomes the problematic issues related to the storage of intermediate and recalcitrant seeds. In theory, it offers indefinite preservation of tissues in liquid nitrogen (LN) vapor (−130 °C) or directly immersed in LN (−196 °C), with cellular metabolism effectively halted and the cessation of cell aging [113,114,115]. Cryostorage is based on the premise that under specific conditions, water can be cooled to cryogenic temperatures in a manner that avoids the process of ice nucleation [39,113,116]. Therefore, a major effort is put into the prevention of ice crystal formation and the promotion of vitrification which can be described as a solidification of a solution by achieving an extremely high viscosity without ice crystallization, resulting in a so-called “glassy” state [117]. In nature, the progressive loss of water inducing such a solid-like state is observed during the maturation of orthodox seeds. As a result, storage at low temperatures for these seeds does not induce any damage, leading to a decline in viability. In contrast, water from seeds with a high cell water content needs to be removed. One way to avoid ice formation is complete drying out of the plant tissue leaving no water, but for obvious reasons, it is not applicable as it would lead to cell death. In the case of desiccation-sensitive seeds, the problem is not trivial, and the solution is usually excision of explants followed by a variety of combined techniques which are: air drying, freeze dehydration, application of penetrating and osmotic active cryoprotectants, and the induction of accumulation of metabolites in cells. Therefore, all cryogenic strategies aim to achieve the glassy state by dealing with two factors: concentration of soluble compounds in the cytoplasm and freezing rate [58,115]. Unfortunately, there are limited studies which aimed to test the viability of seeds and explants after long-term (years and decades) cryogenic storage. However, studies on seeds stored for more than 10 years in cryogenic conditions showed that even though seeds deteriorate faster than anticipated, the shelf life of, i.e., lettuce seeds stored in the vapor and liquid phases of LN was projected for ~500 and ~3400 years, respectively [118]. Another study showed that the initial viability of seeds is an important factor, that needs to be monitored, as short-lived seeds derived from 11 investigated species deteriorated faster during the time of cryostorage (12–20 years) when their initial viability was low [119]. Therefore, there is a continuous need for further research showing how seed viability changes during long-term cryogenic storage, as well as during conventional storage. Special attention should be placed on testing and characterizing the proper storage and recovery procedures, determination of the time of safe storage that does not lower seed viability, and identifying the consequences of any instabilities in stored plant material to ensure the conservation of viable and true-to-type plant germplasm [91,120].

### 5.1. Cryogenic Storage of Orthodox Seeds

Cryogenic storage of orthodox seeds is relatively easy and commonly used in gene banks for ensuring the secure long-term backup of especially valuable seeds; however, it has become a routine practice only in the past 40 years [118]. Prior to cryostorage, seeds are desiccated to the tolerable range of 8–10% MC. Afterwards, they are placed into LN-tolerant containers, i.e., cryovials, or CryoFlex (polyethylene bags, sealed at both ends), and subsequently immersed directly into LN or put into a vapor of LN. The thawing process is also relatively straightforward with seeds being required to be transferred from cryogenic conditions directly to water baths at 40–45 °C. Alternatively, they can be thawed on a laboratory bench at room temperature (15–25 °C). However, a warm water bath is preferred due to the assurance of a more rapid and even thawing, which limits the formation of lethal ice crystals that tend form during a slow thawing procedure. So far, successful cryopreservation protocols have been established for many economically important species or their wild relatives such as *Secale cereale* L. [121], *Solanum lycopersicum* Mill. [122], *Medicago sativa* L., *Beta vulgaris* L., *Allium cepa* L., *Oryza sativa* L., *Zea mays* L. [118]. Orthodox seeds of many tree species can be successfully cryostored as well, such as seeds of *Fraxinus excelsior* L. [123], *Prunus armeniaca* L. [124], *Malus sylvestris* (L.) Mill. [125], *Pyrus communis* (L.) [91,126], *Prunus avium* L. [127,128], *Sorbus aucuparia* L., *P. padus* L., and *Cornus sanguinea* L. [89]. However, even in the case of cryopreservation of orthodox seeds, which tolerate desiccation below 5% of MC, some studies showed that seeds at a MC lower than 5% may produce fewer seedlings after cryostorage [129].

### 5.2. Cryogenic Storage of Intermediate Seeds

For intermediate seeds that do not tolerate conventional storage, cryopreservation can be a potential alternative. However, in this scenario, a single “universal” protocol does not exist, as in the case of orthodox seeds, that enables cryopreservation. As a result, it is important to separately evaluate seeds from every species of interest in order to identify its optimal MC and tolerance to cryogenic storage. However, the number of experimental records demonstrating the successful cryopreservation of intermediate seeds has significantly increased during the past several decades. So far, cryogenic storage for oil-rich seeds as *Coffea liberica* Bull. ex Hiern has been successful at an experimentally indicated optimal MC (16.7%) [130]. Successful cryopreservation was also reported for four *Citrus* species (*C. aurantifolia*, *C. grandis*, *C. madurensis, C. reticulata*) [131]. The second group of intermediate seeds, which is referred to as ‘short-lived seeds’, are also competent for cryopreservation. Successful cryostorage protocols have been published for two *Salix* hybrids: *S. rehderiana* × *S. caprea* and *S.* × *sericans* × *S. viminalis* [132], *S. xerophila*, *S. maximowiczii*, and *S. koreensis* [133], *S. caprea* L. [102], *Populus deltoides* Bartr. [134] and *P. nigra* L. [104]. However, short-lived seeds require permanent monitoring during cryogenic storage because aging processes do not stop completely and have deleterious effects on their longevity [119].

### 5.3. Cryogenic Storage of Recalcitrant Seeds

Entire recalcitrant seeds are not capable of withstanding cryogenic storage, as their tissues do not tolerate desiccation below 35–40% of MC for temperate species and 50–60% of MC for tropical species. Such high hydration of cells promotes the formation of ice crystals during the freezing and thawing of seeds, leading to lethal injuries, in particular to cellular membranes [58]. Even though entire recalcitrant seeds cannot be cryostored, their germplasm can be successfully preserved in LN in the form of tissues isolated from seeds, i.e., embryonic axes (EA) containing both shoot and root meristems [60,135,136,137,138,139,140,141,142] or plumules (containing only shoot meristem). To date, however, only plumules isolated from seeds of *Quercus robur* L., *Q. petraea* (Mattuschka) Liebl. and *Cocos nucifera* L. have been successfully cryostored [112,143,144,145]. The regeneration of plantlets from EA or plumules can be obtained via in vitro culturing. However, even though the EA is often characterized by higher desiccation tolerance than whole seeds [146,147], when isolated from seeds, they do not tolerate cryogenic storage. As a result, additional pre-treatment before plugging in LN is necessary. These processes are mostly vitrification, encapsulation vitrification, and droplet vitrification procedures based upon the utilization of cryoprotectants that are (1) penetrable through the cell wall and into the protoplast, (2) penetrable through the cell wall only, or (3) unable to penetrate through the cell wall [148,149]. For many decades, the most common vitrification solution was plant vitrification solution 2 (PVS2), which contains 30% (*w*/*v*) glycerol, 15% (*w*/*v*) ethylene glycol and 15% (*w*/*v*) dimethyl sulfoxide (DMSO) and 0.4 M sucrose [150]. However, this solution is often modified by additional compounds such as Supercool X1000 [151]. In recent years, cryoprotectants, which lack DMSO, i.e., 50% glycerol, 50% sucrose, have been popular [152], due to the recognition and discussion of the genotoxicity of DMSO [153,154,155].

Cryostorage can be a successful method of securing recalcitrant germplasm derived from species of tropical and temperate zones. However, in many cases, the entire cryopreservation procedure is challenging as it may require several steps including pre-drying, pre-cooling, and the usage of varied vitrification solutions. In addition, it is also required to test the duration of exposure time to cryoprotectants, modification of the pace of drying and cooling in LN, selection of the conditions for optimal sterilization and regeneration during in vitro culture. Collectively, these aforementioned steps and procedures make the cryopreservation technique expensive and labor-consuming in comparison to maintaining deposited seeds in gene banks. Therefore, the cryopreservation methodologies can only be performed in laboratories with specialized equipment.

### 5.4. Cryogenic Storage of Woody Plant Species at Gene Banks

Due to a requirement of a special equipment and training, cryopreservation is not routinely applied in many seed banks. Therefore, cryostorage of seeds of woody plant species is not commonly used, although there are several exceptions. For instance, in Forest Gene Bank in Kostrzyca, there are seed accessions of 20 species of forest trees and shrubs cryopreserved (Appendix A based on [17]). Moreover, to achieve Global Strategy for Plant Conservation Target 8 (CBD 2012), it is necessary to conserve species native to tropical moist forests [156]. Significantly, for many of them, cryopreservation may be the only method to ensure the effective ex situ conservation [157,158]. So far, protocols for cryopreservation of zygotic embryos of tropical recalcitrant seeds of tree species were characterized with various outcome. Some reports showed some success, e.g., cryopreservation of embryonic axes of *Aquilaria malaccensis* Lam., *Sterculia cordata* or *Acrocomia aculeata* (JACQ.) Lodd. ex Mart. [159,160,161]. However, many of reports have informed about failures or unsuccessful attempts of cryostorage of tropical tree species embryos [162,163,164]. So far, only the successful protocol for cryopreservation of *Quercus robur* plumules is routinely applied in gene banks [17,112,144,145]. However, the availability of information about cryopreserved accession in seed banks is very scarce and needs to be updated.

## 6. Future Possibility of Improving Storability of Seeds with Conventional Methods

Improving seed storage procedures via reductions in seed MC and storage temperature has limitations. Consequently, other approaches have been developed in an effort to optimize storage conditions. For example, modulation of the gaseous environment has been evaluated since the aging processes of dry stored seeds is accelerated by the presence of oxygen in the storage environment [165]. Thus, storage of seeds at a low oxygen concentration has been investigated as a method to improve seed viability. Studies which aimed to increase seed longevity of celery and celeriac seeds showed the importance of storage in cool and anoxic or limited oxygen conditions as soon as possible after harvesting and drying [165]. However, it is important to note that storage of seeds under hypoxic conditions also has deleterious effects on viability when moisture levels are relatively high in stored seeds. At high water activity levels, seeds are metabolically active and oxygen deprivation will result in suffocation and anaerobic respiration, accompanied by the production of toxic acetaldehyde and ethanol. Conversely, studies on dry seeds have reported either neutral or positive effects of anoxic storage conditions on seed longevity [165,166]. Recently published studies on the storage of maize seeds showed that seeds stored at 75% and 85% RH and 25 °C benefited from anoxic conditions in terms of their vigor and germination. However, anoxia did not affect seeds when higher temperatures (30 and 35 °C) and 65% RH were investigated [167]. This clearly shows that anoxic conditions benefit seed quality only under particular combinations of MC and temperature, therefore the optimal conditions of storage should be separately investigated for every species. Another recent study showed that modulation of the gaseous environment using oxygen absorbers and/or silica gel have potential for enhancing seed longevity by trapping toxic volatiles emitted by seeds during artificial aging. Moreover, this same study showed that for two investigated species: *Lolium perenne* L. and *Agrostemma githago* L., seed longevity was greater when aged in the presence of silica gel due to its action as a volatile trap. No effects on seed MC or oxygen concentration were observed in the storage containers [168]. It was also shown that 98% of nitrogen in the atmosphere effectively protected the reactive oxygen species (ROS) scavenging systems in wheat seeds and alleviated the internal deterioration during storage of wheat seed at accelerated aging conditions (at 35°C and 86% of RH) [169]. Beneficial effects of nitrogen in the controlled storage atmosphere have been reported for two old Italian wheat cultivars (Verna and Cappelli). Under nitrogen-saturated conditions, the loss of functional molecules, especially vitamin E, was reduced [170]. Moreover, new approaches aiming to improve seed storage include the invigoration of seeds with, i.e., exogenous antioxidants [171,172,173] or usage of nanoparticles for seed protection against microbial infections and alteration of deteriorative physiological processes through ROS scavenging. As a result, strategies incorporating the usage of nanoparticles have the potential for beneficial impact on plant growth, development and stress resistance [7,174,175]. Moreover, recent research focused the attention on the important role of nitric oxide (NO) as a seed antiaging molecule that diminishes the negative effects of seed deterioration processes. Significantly, its role in seed vigor improvement has appeared to be very promising [176,177].

## 7. Future Prospects of Improving the Cryostorage of Seeds or Explants

Cryogenic storage is much more complex than conventional methods, especially when applied for the preservation of recalcitrant embryonic axes or plumules. In both of these latter examples, extraction of these tissues and further treatment requires scientific expertise and specialized equipment. Several parallel technical paths have been developed for cryopreservation of seed materials which depend on the methods of cryo-pretreatment including vitrification, encapsulation vitrification, and droplet vitrification. Since cryopreservation has been more commonly used to preserve shoot tips or embryogenic callus, methods that have been successfully applied for somatic tissues are therefore also utilized for embryonic axes. However, the appropriate design and selection of a suitable cryopreservation protocol is not easy task, especially when information is lacking regarding the cryopreservation of seed explants of closely related species. Therefore, cryopreservation protocols are being constantly modified to achieve the best outcomes, for instance with the application of activated charcoal (AC) to alginate beads. This modification allowed to enhance the post-cryopreservation shoot and root regrowth of AC-encapsulated embryonic axes to *Castanea sativa* Mill. [141]. Another option is the application of exogenous antioxidants at different procedural stages of cryopreservation. The most promising results have been observed with glutathione, ascorbic acid, alpha-tocopherol (vitamin E) and cathodic water. However, most of these components were primarily applied to improve the cryopreservation of shoot tips [178,179] and only cathodic water was successfully applied for the cryopreservation embryonic axes of *Ekebergia capensis* [60]. Recent studies have demonstrated that incorporation of an application of exogenous dehydrin NnRab18 protein improves the cryopreservation of *Arabidopsis thaliana* seedlings [180]. Moreover, it seems that the rapid development of protocols including the application of nanoparticles (i.e., gold, iron oxide, nano-zinc oxide, and selenium), especially in cryopreservation of human and animal cells [181,182,183], shows great promises for future optimizations of plant cryopreservation protocols. Indeed, the first reports of successful application of nanoparticles to improve plant callus or shoot-tip cryopreservation have been reported [184,185].

## 8. Conclusions

The deleterious effects of seed aging occur mostly due to oxidative damage, leading to deterioration of biomolecules. Therefore, the main concept in seed storage is anchored in the prerequisite that effective seed storage relies on slowing down the seed’s normal metabolism, as much as possible without causing damage [31,186]. Consequently, the key to successful post-harvest management and storage of collections of seeds and explants is to understand and control temperature and humidity of storage as well as air accessibility. Obviously, this demands well-equipped storage facilities [31].

Moreover, in addition to the economic importance and conservation efforts, another classification complying with the feasibility and limitations of seed banking was proposed [41]. In this classification, more factors affecting the suitability of seeds to be preserved have been taken into consideration, including their tolerance to desiccation and low temperatures of storage, their accessibility and initial viability, access to dormancy release protocols and germination procedures. This classification has been started from the discrimination of the group of “exceptional” species, which includes species that produce seeds which are ‘un-bankable’ under conventional conditions, species breeding few or no seeds, or for which seed collection is not practical [41]. The concept of exceptional species has been further developed [40,187], and exceptional species were divided into groups based on critical limitations in seed banking. Those limitations are described as exceptionality factors (EF). EF1 distinguishes species that are characterized by seeds that are not viable or infrequent or even not available, EF2 groups recalcitrant seeds, while EF3 corresponds to species producing seeds that partially tolerate desiccation and are short lived at −20 °C, therefore resembling the intermediate category. Finally, EF4 indicates species which produce deeply dormant seeds. Since there are fewer exceptional species than species producing bankable seeds, and their storage requires skills and technologies that are far more complex and expensive than conventional seed banking, they are therefore far more likely to be overlooked. However, this classification may be commonly used in the future, particularly by practicians in gene banks as the identification of a species as exceptional is a first step in developing targeted, effective, and scientifically informed species conservation strategies [40].

Even though we have significantly improved our knowledge about seed long-term storage over the last 50 years, there are many issues that still need to addressed. Firstly, for successful seed conservation, more attention should be placed into the development of germination and dormancy breaking protocols, as well as the determination of seed post-harvest physiology, and finally into the testing of long-term storage of seeds (years/decades). Because improvement in this area requires labor-consuming research and time that usually goes beyond the 3–4-year cycle of research funding, it is very difficult to reach solid experimental conclusions within the time frames required by financing agencies. However, an experiment conducted at Michigan State University [28] shows that seed science is a marathon, not a sprint. Therefore, new funding opportunities, taking into account long-term research pursuits, should be provided.

Cryobiotechnology, including cryopreservation and in vitro technologies, provides potential technical paths to preserve plant species for which seed banking is not an option, especially sub-tropical and tropical recalcitrant-seeded species. For example, a recent study demonstrated that the viability of plant tissues can be maintained over 20–30 years in cryogenic conditions [34,37]. Finally, it is crucial to ensure proper storage conditions that do not exert any deleterious impacts on seedling development. Therefore, in addition to the assessment of germination, other seed and seedling properties should also be more commonly tested such as morphological properties [188], proteome and metabolome status [98], and genetic and epigenetic stability [96,120,145,189,190]. Consequently, there is a huge need for new biochemical or molecular markers of seed viability that improve the process of monitoring plant material during storage in gene banks and to help with the understanding of the concomitant aging process occurring in seeds. These studies have developed rapidly over the last decade and substantiate the importance of volatile organic compounds [94,95,98], proteomic studies [191], RNA integrity [192,193,194,195,196] and DNA damage markers or epigenetic marks [96,97,99,190,197,198,199,200] that are highly promising and should be more widely explored.

## Figures and Tables

**Figure 1 plants-12-00072-f001:**
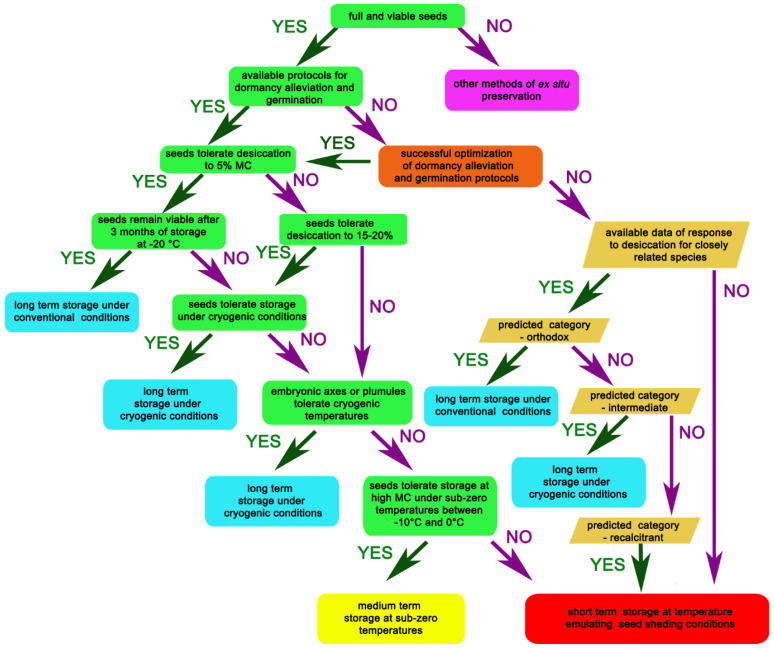
A protocol to determine the optimal conditions of plant germplasm storage. Two paths are presented: experimental in rectangles and literature based in parallelograms.

## Data Availability

Not applicable.

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
