# Peer review of "Choosing the Right Path for the Successful Storage of Seeds"

_plants, 2022, doi:10.3390/plants12010072_

Round 1

Reviewer 1 Report

Revision of the --Manuscript Draft-- Manuscript Number: plants-2044787

Title: Choosing the right path for the successful storage of seeds.

Type: Reviews

This is a manuscript within the scope of the journal "Plants" and it is presented in a well-structured manner. The manuscript is scientifically sound. The review is clear, comprehensive and of relevance for successful illustration of the scheme that can be used to ensure the most optimal conditions to enable the long-term storage of seeds. Moreover, the review provides an advancement of the current knowledge to confirm the main questions of seeds cold storage at −20 °C. The manuscript is very important to fully understand the capability or lack thereof for seeds or their ex-plants to tolerate cryogenic conditions too.

The cited total references are 182. The number of self-citations is 17. The recent publications within the last five years are 75 and represent 41.2% of the total references: in 2022 (19 references); in 2021 (19 references); in 2020 (23 references); in 2019 (9 references) and in 2018 (5 references). The referred publications are relevant according to the Web of Science Citation Index Expanded (SCIE) and SCImago Journal Rank (SJR (Q1)) impacts. For example, Plants, BMC Plant Biology, Planta, Physiologia Plantarum, Journal of Experimental Botany, Frontiers in Plant Science, Plant Biology, Nature Plants are relevant in  PLANT SCIENCES – SCIE(Q1); Biodiversity and Conservation in BIODIVERSITY CONSERVATION - SCIE(Q1); Forests, Tree Physiology in FORESTRY - SCIE(Q1); Proceedings of the National Academy of Sciences in MULTIDISCIPLINARY SCIENCES - SCIE(Q1); Animals in AGRICULTURE, DAIRY & ANIMAL SCIENCE - SCIE(Q1); VETERINARY SCIENCES - SCIE(Q1); Plants People Planet in PLANT SCIENCES - SCIE(Q1); BIODIVERSITY CONSERVATION - SCIE(Q1); ECOLOGY - SCIE(Q1); Plant Methods in BIOCHEMICAL RESEARCH METHODS - SCIE(Q1); PLANT SCIENCES - SCIE(Q1); Conservation Biology in ECOLOGY - SCIE(Q1); ENVIRONMENTAL SCIENCES - SCIE(Q1); BIODIVERSITY CONSERVATION - SCIE(Q1).

From my point of view, this current review is relevant but the interest to the scientific community could enhance if the authors take in consideration the following topics:

In the second point (2. Important questions that need to be addressed prior to seed storage) I suggest to add one paragraph of the key concepts in seed storage and the steps to take for effective storage of native seeds for restoration use, that appeared in the reference: De Vitis, M., Hay, F. R., Dickie, J. B., Trivedi, C., Choi, J., & Fiegener, R. (2020). Seed storage: maintaining seed viability and vigor for restoration use. Restoration Ecology, 28, S249-S255.

I suggest reading the publication: Waterworth, W. M., Bray, C. M., & West, C. E. (2019). Seeds and the art of genome maintenance. Frontiers in Plant Science, 10, 706. In this paper the authors mentioned that an increasing number of studies are revealing DNA damage accumulated in the embryo genome, and the repair capacity of the seed to reverse this damage, as major factors that determine seed vigor and viability. This aspect is not explicit in the actual manuscript.

One gap in the review is related with the effect of Nitrogen Reactive compounds on aging in seed and the relevance of Nitric Oxide for plant physiology during every ontogenetic stage from seed germination to plant senescence. I suggest developing this topic and read the publication: Ciacka, K., Staszek, P., Sobczynska, K., Krasuska, U., & Gniazdowska, A. (2022). Nitric Oxide in Seed Biology. International Journal of Molecular Sciences, 23(23), 14951.

However, according to the revised manuscript the minor issues are related with the following topics:

 One problem of the current manuscript is the not appropriate use of Abbreviations for the whole document. Please, consider the suggestions in the attached document e.g., LNs 119, 156, 162, 168, 212, 213, 432, 433, 484, 491, 493,

Moreover, I suggest the authors for improving the manuscript to take care with details that need to be accurately corrected for the whole document. See in the attached document the LNs 37, 67, 68, 100, 126, 204, 207, 210, 211, 214, 216, 233, 271, 293, 295, 296, 320, 387, 391, 405, 408-410, 466, 526, 548,

Author Response

Response to Reviewer 1

We addressed all comments provided by Reviewer 1 in the letter and directly in the manuscript’s commentary section. Three references were introduced into manuscript with short comments.  We would like to thank the Reviewer for the contribution to perfecting the manuscript. We hope to get the Reviewer’s approval. 

Reviewer 2 Report

The review about paths for seed storage provides information about different ways for optimal storage; information is given about conventional and cryogenic storage of orthodox, intermediate and recalcitrant seeds. The review gives a good overview of successful storage options.

The paper tries to cover all plants. I wonder why woody plants have an own chapter. Also, in the other chapters, woody plants are mentioned. This is not logic and should be corrected. Otherwise, we would need also differentiation between more types like aquatic plants, crops, wild plants, different climatic zones, etc.

In addition, it should be shortly mentioned that we have beside the different seed types intraspecific variability.

Author Response

Response to Reviewer 2

We have submitted our review to Plants’ special issues: Application of Biotechnology to Woody Propagation. Therefore, in our manuscript we have put special attention on woody plants and storage of their germplasm. Also, we have acknowledged Editor’s suggestion and we have prepared two additional chapters entitled: “Conventional storage of woody plant species in gene banks” and “Cryogenic storage of woody plant species at gene banks”. Consequently, the current version of our manuscript is prepared to meet the requirements of the special issue. We hope to get the Reviewer’s approval based on that clarification.

Reviewer 3 Report

Considering the complexity of these topics, which probably deserve to be written in a big Book, I found this Review interesting. Of course, many topics could be covered in more detail, but I understand the difficulties in deepening all the cases.

I suggest some minor revisions

LLs 22-37: Why do you speak only of global warming? There are many other threats impacting heavily the loss of biodiversity.

LLs 47-52: This is a Review; please upload these information adding also some recent published paper.

LLs 53-56: Many other efforts have been made for the ex situ conservation of plant biodiversity in the world. Could it be possible to cite others here?

LLs 57-60: You are citing here papers dated 1926 and 1967. Are you sure that do not exist more recent work about these topics?

LL 205: Are you sure "50°C"?

LLs 258-259: Several Germplasm Banks store their seeds for long term at -25°C, in particular for wild Mediterranean species; see for example (http://dx.doi.org/10.7338/pls2017542S1/11) or another germplasm banks of Ribes Network (https://www.reteribes.it/index.asp), GENMEDA (http://www.genmeda.net/) or ENSCONET consortium.

LLs 278-279: Apiaceae, Brassicaceae, Malvaceae and Chenopodiaceae no italics.

Author Response

Response to Reviewer 3

We addressed all comments provided by Reviewer 3 in the letter. Additional references related to current state-of-the-art in germplasm storage have been added, as well as references indicating other reasons of biodiversity decline (in the Introduction chapter). We would like to thank the Reviewer for all suggestions. We hope to get the Reviewer’s approval. 

Reviewer’s comment: LLs 57-60: You are citing here papers dated 1926 and 1967. Are you sure that do not exist more recent work about these topics?

Our response: I our review we have tried to show very new as well as former but still valid references.

Reviewer’s comment: LL 205: Are you sure "50°C"?

Our response: We confirm that the value is correct.

Reviewer’s comment: LLs 258-259: Several Germplasm Banks store their seeds for long term at -25°C, in particular for wild Mediterranean species; see for example (http://dx.doi.org/10.7338/pls2017542S1/11) or another germplasm banks of Ribes Network (https://www.reteribes.it/index.asp), GENMEDA (http://www.genmeda.net/) or ENSCONET consortium.

Our response:  References have been included.

Reviewer’s comment: LLs 278-279: Apiaceae, Brassicaceae, Malvaceae and Chenopodiaceae no italics.

Our response: Corrected.